# Floating Active Inductor Based Trans-Impedance Amplifier in 0.18 μm CMOS Technology for Optical Applications

**Xiangyu Chen [1],* and Yasuhiro Takahashi [2]**

[1]   Graduate School of Engineering, Gifu University, 1-1 Yanagido, Gifu-shi 501-1193, Japan
[2]   Department of EECE, Faculty of Engineering, Gifu University, 1-1 Yanagido,
      Gifu-shi 501-1193, Japan; yasut@gifu-u.ac.jp
*   Correspondence: xiangyuchenCN@hotmail.com; Tel.: +81-58-293-2692

**Abstract:** In this paper, a transimpedance amplifier (TIA) based on floating active inductors (FAI) is presented. Compared with conventional TIAs, the proposed TIA has the advantages of a wider bandwidth, lower power dissipation, and smaller chip area. The schematics and characteristics of the FAI circuit are explained. Moreover, the proposed TIA employs the combination of capacitive degeneration, the broadband matching network, and the regulated cascode input stage to enhance the bandwidth and gain. This turns the TIA design into a fifth-order low pass filter with Butterworth response. The TIA is implemented using 0.18 μm Rohm CMOS technology and consumes only 10.7 mW with a supply voltage of 1.8 V. When used with a 150 fF photodiode capacitance, it exhibits the following characteristics: gain of 41 dBΩ and −3 dB frequency of 10 GHz. This TIA occupies an area of 180 μm × 118 μm.

**Keywords:** transimpedance amplifier (TIA); active inductor (AI); chip area; bandwidth enhancement

---

## 1. Introduction

With the rapid increase in the amount of data transmitted over telecommunications networks, interest in high speed optoelectronic devices and systems has increased. The high demand for large data transmission rates has led to the rapid development of data transmission technology. With the continuous increase of data traffic in modern communication networks, the data rate between system nodes has approached the physical limit of general links such as copper wires. Therefore, an urgent need is to increase the communication speeds of these networks. Owing to its high bandwidth, the optical fiber is the favorite medium chosen to meet this need in today's information based society [1–4]. Our research efforts are aimed in this direction [5–7].

The recent dramatic growth of Internet data volume and speed entails the development of low cost integrated optical communication systems with a high transmission bandwidth [8]. The analog optical receiver that incorporates a TIA input stage and post-amplifying stages plays a critical role in the whole optical communication system [9]. The TIA is the first electrical building block in the analog optical receiver. Its function is to convert the induced photodiode (PD) current into a large voltage signal to be used in the digital processing unit. The TIA is required to have both high gain and wide bandwidth along with low power dissipation [10–33].

Several research articles on improving the bandwidth of TIA have been published. In particular, inductive peaking has been extensively used to improve the bandwidth and decrease parasitic capacitance

effects. The inductor is one of the most important elements in TIAs for high frequency applications and RF systems. However, its extremely large size makes the TIA chip large and expensive. Hence, decreasing the area of the inductors used in TIA design is very important. In the gyrator realization, the active inductor (AI) is constructed by connecting two differential transconductors back-to-back in negative feedback. The parasitic capacitance of the basis transistors is the load capacitance in the gyrator-C topology.

Furthermore, a bandwidth enhancement method based on the interaction of a matching LC network, a regulated cascode (RGC) input stage, and a capacitive degeneration stage can be used to enhance the bandwidth and provide efficient noise reduction. This turns the TIA into a fifth-order Butterworth low pass filter. In this paper, we propose a novel wideband TIA using the Mahmoudi–Salama floating-type AI, which is based on the gyrator-C topology.

This proposed TIA based on floating AI not only has the advantages of a wide bandwidth of 10 GHz and a transimpedance gain of 41 dBΩ, it also has the merits of a smaller chip area and lower power dissipation compared with the conventional TIA, based on a spiral inductor [8]. However, owing to the use of floating active inductors, this proposed TIA has the inherent disadvantage of reduced input referred noise performance in comparison to the conventional TIA, made of a spiral inductor [8]. In this paper, the design and post-layout simulation results of a 10 GHz 0.18 µm CMOS TIA are presented.

The remainder of this paper is organized as follows. Section 2 describes the conventional TIA based on the traditional broadband design technique. Section 3 discusses the basic principle of gyrator-C networks and introduces the gyrator based differential FAI used to replace the spiral inductor in the conventional TIA. The proposed TIA based on the FAI is presented in Section 4. The simulation results are presented in Section 5, and finally, the conclusions are summarized in Section 6.

## 2. TIA Based on the Traditional Broadband Design Technique

Figure 1 represents the schematic of a TIA based on the traditional broadband design technology. The TIA was composed of the following four parts: a matching network, an RGC input stage, a gain stage with capacitive degeneration, and a source follower output stage [8]. Because the input impedance of the RGC stage was very small, the lowest pole of the circuit was located within the TIA. In this TIA, a small $R_2$ was chosen to avoid possible peaking due to the zero generated by the local feedback of the RGC stage. Moreover, relatively large $R_s$ were selected to minimize the noise current and signal loss. The capacitive degeneration gain stage consisted of $M_3$, $R_3$, and $R_b$, and $C_b$ contributed a zero $(R_b C_b)^{-1}$ that was used to compensate the lowest pole determined. $M_4$ and $R_4$ formed the source follower output stage used to drive the capacitance of the output pad.

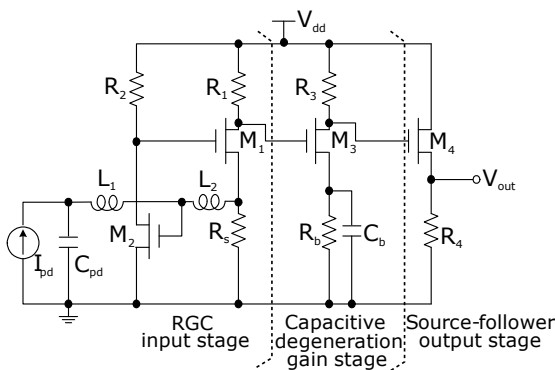

**Figure 1.** Schematic of a TIA based on traditional broadband design technology.

## 3. Lossless Floating Gyrator-C Active Inductors

### 3.1. Principles of Gyrator-C Active Inductors

The inductor is one of the most important circuit elements in high frequency applications and RF systems. CMOS AIs have become a very hot research topic in recent years due to their many advantages over passive inductors. AI systems can be implemented by circuits that occupy a small chip area. In addition, AIs have large variable inductance and a high quality factor. In contrast, compared with spiral inductors, AIs have some disadvantages such as poor noise performance, small dynamic range, and sensitivity to the process. Bandwidth extension topologies typically use on-chip inductors to compensate for capacitive effects, but the associated hardware costs are high. Therefore, it is very important to reduce the area of the inductor used in TIA design.

An AI was constructed with a small number of transistors. The gyrator was implemented using two transconductances connected back-to-back [18]. When the port of the gyrator is connected to a capacitor, as shown in Figure 2, the network is called the gyrator-C network. A gyrator-C network is said to be lossless when both the input and output impedances of the system's transconductor of the system are infinite, and the transconductances of the transconductors are constant.

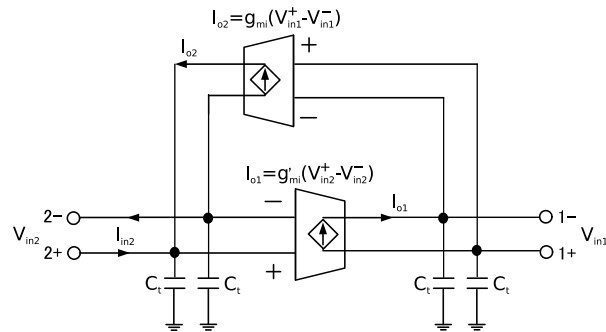

**Figure 2.** Schematic of lossless floating gyrator-Cactive inductors.

The admittance looking into port 2 of the gyrator-C network is given by:

$$
\begin{aligned}
Y &= \frac{\frac{2g_{mi}g'_{mi}}{sC_t}\left(V^+_{in2} - V^-_{in2}\right)}{V^+_{in2} - V^-_{in2}} \\
&= \frac{2g_{mi}g'_{mi}}{sC_t} \\
&= \frac{1}{s\left(\dfrac{\frac{C_t}{2}}{2g_{mi}g'_{mi}}\right)}.
\end{aligned}
\tag{1}
$$

Equation (1) indicates that the port 2 of the gyrator-C network behaves as a floating inductor with inductance given by:

$$
L = \frac{\frac{C_t}{2}}{g_{mi}g'_{mi}}.
\tag{2}
$$

## 3.2. Mahmoudi and Salama Active Inductor

The floating active inductor proposed by Mahmoudi and Salama is used in the design of quadrature down converters for wireless applications [15,21]. The schematic of the the floating active inductor is shown in Figure 3. It mainly consisted of a pair of differential transconductors and a pair of negative resistors at the output of the transconductors. $M_8$ and $M_{16}$ were biased in the triode and behaved as voltage controlled resistors. They were added to the conventional cross-coupled configuration of negative resistors to provide resistance tunability for the negative resistors, without using a tail current source. The common-mode stabilizer consisted of a cross-coupled differential pair $M_6$, $M_7$ ($M_{14}$, $M_{15}$) and transistor $M_8$ ($M_{16}$), which was designed to operate in triode mode. The stabilizer was added to the design to stabilize the common-mode behavior of the differential active inductor by moving the common-mode right half-plane pole of the inductor, which may cause instability, to the left-half-plane. In the differential mode of operation, the common-mode stabilizer appeared as an impedance $Z_o$ with a real part given by [34]:

$$\text{Re}\{Z_o\} = -\frac{1}{g_{ms}}||r_{os},\tag{3}$$

where $g_{ms}$ is the transconductance of $M_6$ or $M_7$ and $r_{os}$ is the output resistance of $M_8$ in the triode region, which can be controlled by the voltage $V_{b3}$. Hence, the stability of the active inductor was improved.

Figure 4 shows clearly that the Mahmoudi–Salama AI was based on the gyrator-C structure. According to the inductance relationship of (2), when the gyrator-C network appeared as a FAI described in Section 3.1, we could obtain:

$$L = \frac{C}{g_{mi}g'_{mi}},\tag{4}$$

where $g_{mi}$ and $g'_{mi}$ are the transconductances of $M_{10}$ ($M_{11}$) and $M_2$ ($M_3$), respectively.

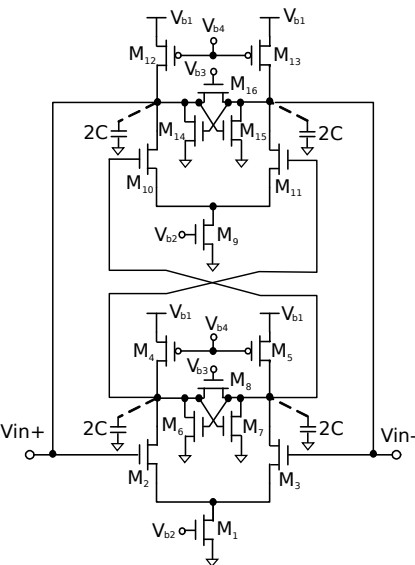

**Figure 3.** Simplified schematic of the Mahmoudi–Salama active inductor.

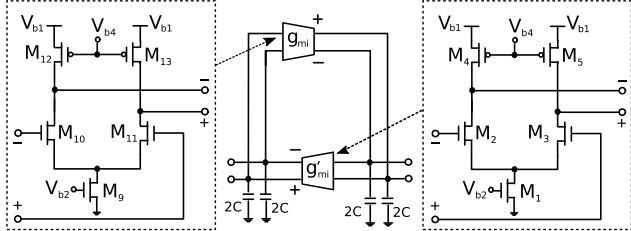

**Figure 4.** The principle diagram of the Mahmoudi–Salama differential active inductor.

The frequency response of the Mahmoudi and Salama AI as obtained through simulation is shown in Figure 5. We can see that the FAI had a good inductance characteristic when the frequency was within 100 GHz.

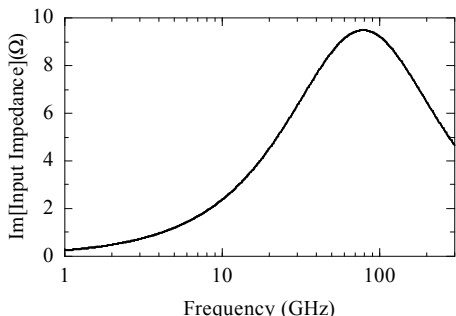

**Figure 5.** Frequency response of the Mahmoudi and Salama active inductor.

## 4. Proposed TIA

### 4.1. Circuit Topology

The RGC input stage was well suited for broadband TIA design by its very low input impedance, compared with a single common-source follower and an inverter based amplifier. Thus, in this paper, the proposed TIA was based on an RGC TIA, as shown in Figure 1. To reduce the chip area of TIA, an inductor $L_2$ was composed by an active inductor, which was named the Mahmoudi and Salama active inductor. Due to its good stability, we chose this type of active inductor (see Section 3.2).

### 4.2. Circuit Structure

The conventional TIA is shown in Figure 1, where $L_1$ and $L_2$ are spiral inductors. In this work, we changed spiral inductor $L_2$ with the Mahmoudi–Salama FAI. Unlike the conventional TIA in [1], in this work, we assumed that $L_1$ was constructed by bonding wires. It is well known that in general processes, the inductance of a 1 mm bonding wire is between 2 nH and 2.5 nH. Therefore, we assumed that the bonding wire, not on-chip, was the inductor $L_1$, as shown in Figure 1. In Section 5, we provide post-layout simulation results that indicated that the value of $L_1$ was between 2 nH and 3 nH.

As mentioned in Section 2, the proposed TIA was composed of four parts: a matching network, an RGC input stage, a gain stage with capacitive degeneration, and a source follower output stage.

### 4.3. Small Signal Analysis of the RGC Stage in TIA

Figure 6 illustrates the small signal model of the matching network and the RGC input stage. As described in Section 2, in the RGC stage of the proposed TIA, we chose a small $R_2$ to avoid possible peaking due to the zero generated by the local feedback of the RGC stage. Here, through small signal analysis, we discuss the effects of the inductors $L_1$, $L_2$, and $R_s$ on the proposed TIA.

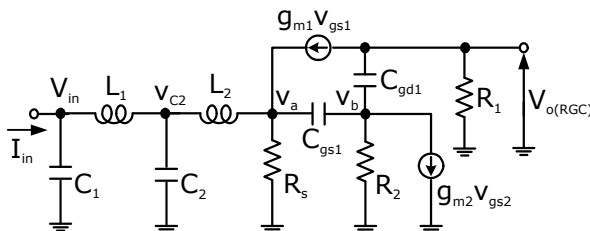

**Figure 6.** Small signal model of the matching network and the RGC stage.

Here, the input impedance $Z_{in}$ can be obtained as:

$$Z_{in(s)} = \frac{v_{in}}{i_{in}} = \frac{sL_1(1 - C_2L_1)}{(1 - C_1L_1)(1 - C_2L_1) - 1}.$$ (5)

$$\left[\frac{1}{sL_1} - \frac{1}{1 - L_1C_2}\left(\frac{1}{sL_1} + sC_2 + g_{m2} + \frac{1}{sL_2R_S} + \frac{sR_sC_{gs1}}{1 + sR_2C_{gs1}}\right)\right]v_{in} = \frac{v_{o,RGC}}{R_1}$$ (6)

$$\begin{aligned} Z_T(s) = \frac{v_{o,RGC}}{i_{in}} &= R_1\left[\frac{1}{sL_1} - \frac{1}{1 - L_1C_2}\left(\frac{1}{sL_1} + sC_2 + g_{m2} + \frac{1}{sL_2R_s} + \frac{sR_sC_{gs1}}{1 + sR_2C_{gs1}}\right)\right]\frac{sL_1(1 - L_1C_2)}{(1 - L_1C_1)(1 - L_1C_2) - 1} \\ &= R_1\left[1 - \frac{1}{1 - L_1C_2}\left(1 - L_2C_2 + sg_{m2}L_1 + \frac{L_1}{L_2R_s} - \frac{R_sL_1C_{gs1}}{1 + sR_2C_{gs1}}\right)\right]\frac{1 - L_1C_2}{(1 - L_1C_1)(1 - L_1C_2) - 1} \end{aligned}$$ (7)

$$\begin{aligned} Z_T(0) &= R_1\left[1 - \frac{1}{1 - L_1C_2}\left(1 - L_2C_2 + \frac{L_1}{L_2R_s} - R_sL_1C_{gs1}\right)\right]\frac{1 - L_1C_2}{(1 - L_1C_1)(1 - L_1C_2) - 1} \\ &\approx R_1\left[1 - \frac{1}{(1 - L_1C_1)(1 - L_1C_2) - 1}\left(1 - L_2C_2 + \frac{L_1}{L_2R_s} - R_sL_1C_{gs1}\right)\right] \end{aligned}$$ (8)

From Equations (5) and (6), the expression of the transimpedance gain $Z_T(s)$ is obtained as shown in (7). In addition, we have the DC gain of the RGC as shown in Equation (8).

The original RGC gain was $Z_T(0)$, as shown in (8). In order to make the gain as large as possible, the term of $\frac{L_1}{L_2R_s}$ must be increased, that is the value of $L_1$ needs to be increased or the value of $L_2R_s$ needs to be decreased. However, in this TIA, $L_1$ was the inductance of the bonding wire, which was almost fixed. As can be seen from Figure 5, the inductance of this FAI was very small. Therefore, we increased $R_s$ above 500 Ω to increase the gain appropriately. Furthermore, a relatively large $R_s$ could minimize noise current contribution and the signal loss due to it.

### 4.4. Noise Analysis

The TIA model with series inductive matching between the photodiode and the amplifier is renowned to be very helpful in reducing the frequency dependent noise and improving the front-end

sensitivity [8,35]. In [8], the equivalent input noise current spectral density of RGC TIA with inductor peaking was approximated as:

$$|i_{n,eq}|^2 = (1 - \omega^2 L_1 C_{pd})^2 I_n^2 + \omega^2 C_{pd}^2 E_n^2. \tag{9}$$

where $C_{pd}$ stands for the photodiode parasitic capacitance. For a given $E_n - I_n$ noise model (see [8]), $E_n$ and $I_n$ are independent of the input matching network. Since $L_1$ is a negative term in (9), the noise reduction effect of $L_1$ can be clearly indicated. The effect of $L_2$ is similar to that of $L_1$. However, the effective inductance of $L_2$ is $L_{2,eff} = \frac{L_2}{(1+g_{m2}R_2)}$, which is relatively small; hence, the noise reduction effect of $L_2$ should be less than that of $L_1$ [8]. Consequently, we can come to the conclusion that increasing the effective value of the inductor in this topology could reduce the input referred noise, in theory.

## 5. Post-Layout Simulation Results

To evaluate its performance, the TIA was implemented using 0.18 µm Rohm CMOS technology. All post-layout simulation results were performed in Cadence.

The layout of the top cell with the pad and TIA core is shown in Figures 7 and 8, respectively. Correspondingly, they occupied layout areas of 1590 µm × 780 µm and 180 µm × 118 µm. Figure 9 shows the chip microphotograph of the proposed TIA.

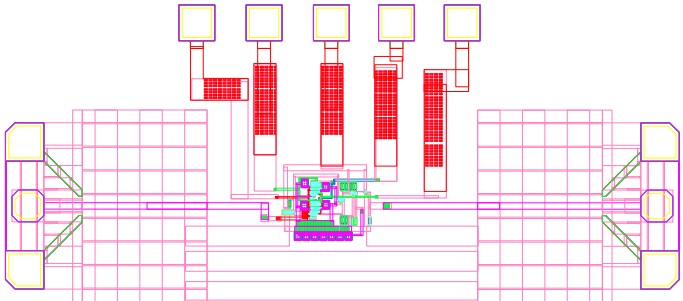

**Figure 7.** Layout of the proposed TIA (top cell with pad).

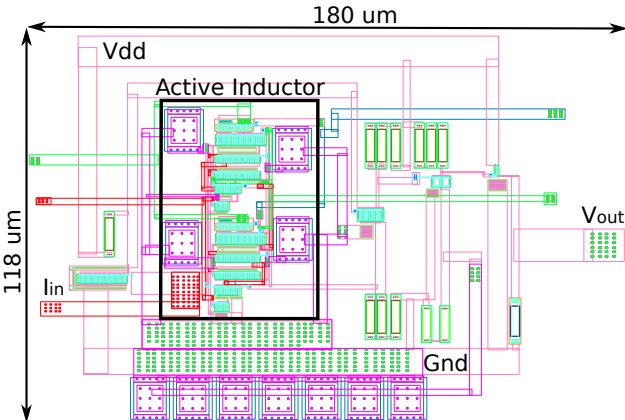

**Figure 8.** Layout of the proposed TIA (TIA core).

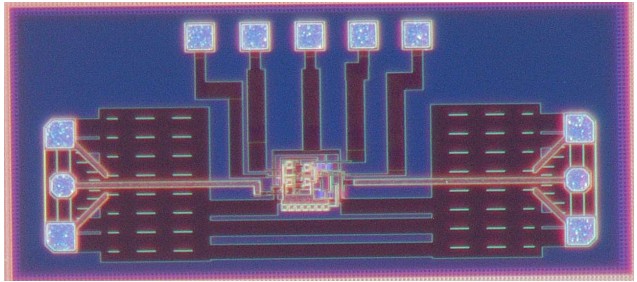

**Figure 9.** Chip microphotograph of the proposed TIA design.

Figure 10 shows the post-layout frequency response of the proposed TIA. We can see from the simulation results that no matter what value $L_1$ took between 2 nH and 3 nH, the transimpedance gain was about 41 dBΩ, and the −3 dB bandwidth was greater than 10 GHz in the presence of a $C_{pd}$ of 150 fF. The conclusion was that any value between 2 nH and 3 nH of the bonding wire worked well with this design and gave good frequency characteristics.

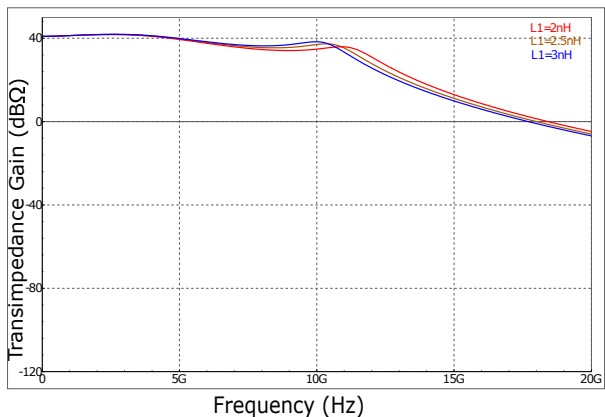

**Figure 10.** Post-layout simulated frequency response of the proposed TIA.

Figure 11 shows the eye-diagram for the input signal currents of 100 $\mu A_{pp}$ $2^{31} − 1$ pseudo-random bit sequence (PRBS) at the rates of 5 Gb/s, 10 Gb/s, and 15 Gb/s. In this post-layout simulation, the PRBS generator was implemented using linear feed back shift registers (LFSR). These PRBS outputs had a jitter of 2 ps. From the post-layout simulation results, the proposed TIA could generate the waveform with good eye-opening owing to the wide bandwidth. The jitter of the post-layout simulation was 4.03 ps when the bit rate was 15 Gb/s.

Table 1 compares the performance of the proposed TIA with other recently published TIAs. It can be clearly seen that the presented work was superior in terms of bandwidth, power dissipation, and chip area compared with other TIAs implemented using the same 0.18 μm CMOS technology.

The standard figure of merit (FoM) is calculated as (10) below and is used to compare this design with other recent TIA designs in Table 1.

$$\text{FoM} = \frac{\text{Gain(dBΩ)} \times \text{Bandwidth(GHz)}}{\text{Power(mW)} \times \text{Chip Area (mm}^2\text{)}} \tag{10}$$

It can be seen that this design was better in terms of FoM compared with the conventional design using the same process. The input referred noise of the proposed TIA was increased by a factor of approximately 1.7, compared with the conventional TIA. The main cause of the increase of input referred noise was that the effective value of the active inductor we used was much smaller than the effective value of the spiral inductor in the conventional TIA. Furthermore, we assumed that the bonding wire $L_1$ was 2 nH to connect the photo-diode (PD). However, in the conventional TIA, the spiral inductor of $L_1$ was 0.77 nH, which was not including the inductance of PD connection. The actual value of $L_1$ in the conventional TIA could be approximately expressed as $L_1(\text{Actual}) = L_1 + L(\text{PD}_{\text{Connection}})$. In other words, assuming that the inductance of the bonding wire was the same, the effective value of $L_1$ of the conventional TIA was actually 0.77 nH larger than that of the proposed TIA. This also increased the input referred noise to a considerable extent. Nevertheless, the noise performance of the proposed TIA was also better than that of the general TIA without a matching network of inductors.

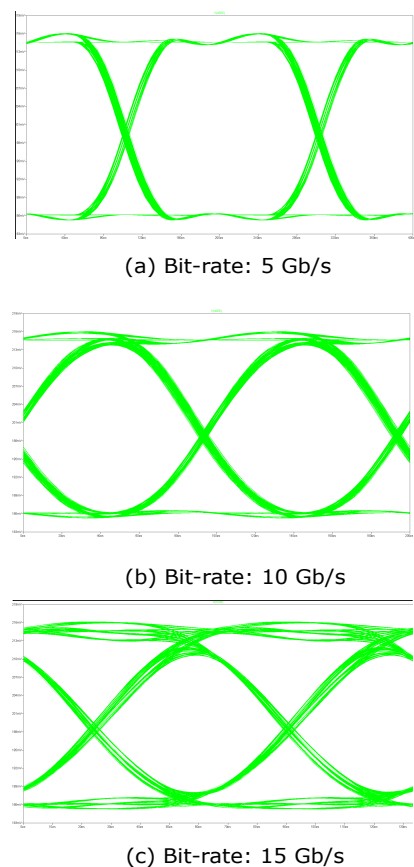

(a) Bit-rate: 5 Gb/s

(b) Bit-rate: 10 Gb/s

(c) Bit-rate: 15 Gb/s

**Figure 11.** Eye-diagram characteristics with a $2^{31} - 1$ pseudo-random bit sequence (PRBS) input current of 100 $\mu A_{pp}$ at (**a**) 5 Gb/s, (**b**) 10 Gb/s, and (**c**) 15 Gb/s.

Table 1. Performance summary and comparison with other works.

| Reference | [8] | [20] | [17] |
|---|---|---|---|
| Technology | 180 nm CMOS | 180 nm CMOS | 180 nm CMOS |
| Topology | RGC + Inductor Peaking | RGC + Inductor Peaking | CS + Inductor Peaking |
| Supply Voltage | 1.8 V | 1.8 V | 1.8 V |
| Transimpedance Gain (dBΩ) | 53 | 55 | 51 |
| Bandwidth (GHz) | 8 | 7 | 30.5 |
| $C_{pd}$ (fF) | 250 | 200 | 50 |
| Power Dissipation (mW) | 13.5 | 18.6 | 60.1 |
| Input Referred Noise (pA/$\sqrt{Hz}$) | 18 | 17.5 | 55.7 |
| Chip Area (μm$^2$) | 450 × 250 | 400 × 250 | 1170 × 460 |
| FoM | 283 | 207 | 48 |
| Results | Measured | Measured | Measured |
| | [26] | [19] | [12] |
| | 180 nm CMOS | 180 nm CMOS | 80 nm CMOS |
| | CS + NI + AI | RGC + AI | CG + Inductor Peaking |
| | 1.8 V | 1.8 V | 1 V |
| | 54.3 | 56 | 52 |
| | 7 | 8.27 | 20 |
| | 50 | 300 | 100 |
| | 29 | 35 | 2.2 |
| | 5.9 | 20 | <50 |
| | 230 × 45 | 106 × 100 | 140 × 70 (* 49,613 μm$^2$) |
| | 1266 | 1202 | 9528 |
| | SPICE | Post-layout | Measured |
| | [22] | [36] | **This work** |
| | 40 nm CMOS | 28 nm CMOS | 180 nm CMOS |
| | Inverter + CD | CG + AI | RGC + AI |
| | 1.1 V | 1 V | 1.8 V |
| | 47 | 43 | 41 |
| | 8 | 22 | 10 |
| | 250 | 150 | 150 |
| | 2 | 2 | 10.7 |
| | 23 | N/A | 30.7 |
| | 200 μm$^2$ (* 4050 μm$^2$) | 18 × 23 (* 17,110 μm$^2$) | 180 × 118 |
| | 46,419 | 27,644 | 1804 |
| | Post-layout | Measured | Post-layout |

CS: common source, NI: negative impedance, AI: active inductor, CG: common gate, CD: common drain, * scaled chip area by Area (scaled) = Area (actual) × $\left(\frac{180(\text{nm})}{\text{Process(nm)}}\right)^2$.

## 6. Conclusions

In this paper, we presented a broadband TIA with a floating active inductor exhibiting the following characteristics: a −3 dB bandwidth greater than 10 GHz and a transimpedance gain of 41 dBΩ. Owing to the use of FAI, the area of the chip was greatly reduced and was almost 18.8% of that of the conventional TIA. Moreover, because the parameters and resistance value of the transistor were different from the conventional TIA, the simulation results showed that the floating active inductor did not increase the power

dissipation. In contrast, the proposed TIA had a lower power dissipation of 10.7 mW. Thus, the post-layout simulation results indicated that the floating active inductor was very useful in optical applications.

**Author Contributions:** X.C. and Y.T. contributed to the design of the proposed circuit; X.C. performed the simulations, the layout design, and the writing, original draft; all authors participated in the data analysis and paper revision.

**Funding:** This research was funded by China Scholarship Council (CSC) (No. 201808050092).

**Acknowledgments:** This work was supported by the VLSIDesign and Education Center (VDEC), the University of Tokyo, in collaboration with Cadence Corporation and Synopsys, Inc. The VLSI chip in this study was fabricated in the chip fabrication program of VDEC, the University of Tokyo, in collaboration with Rohm Corporation and Toppan Printing Corporation. We would also like to acknowledge the financial support by the China Scholarship Council (CSC) through the award of a Ph.D. scholarship (No. 201808050092) to Xiangyu Chen, which is under the State Scholarship Fund.

**Conflicts of Interest:** The authors declare no conflict of interest.

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
