# Peer review of "Floating Active Inductor Based Trans-Impedance Amplifier in 0.18 μm CMOS Technology for Optical Applications"

_electronics, doi:10.3390/electronics8121547_

Round 1
Reviewer 1 Report
Generally, this proposed article for TIA with FAI structure is good. The simulation consequences in chip area, power consumption, gain, BW, PRBS, and FoM compared with the traditional TIA are impressive. If the TIA in your article can be verified on wafer and measured, which show the approached consequences between simulation and measurement, I am glad to recommend it to be published in this great journal. Recently, the content in this article is hard to convince the readers to fully accept your ideas.
Author Response
Dear reviewer
We appreciate your comments on this article and have carefully revised it in accordance with your suggestions. The responses to your comments and questions are shown in the attachment.
Sincerely,
Xiangyu Chen and Yasuhiro Takahashi

Reviewer 2 Report
This paper presents the Floating Active Inductor Based Trans-impedance Amplifier in 0.18 μm CMOS Technology for Optical Applications. Only one concern is listed as follows.
Could you provide the measured results? Or highlight the results from measured or simulation in the comparison table (Table I).
Author Response
Dear reviewer
Thank you very much for taking time out of your busy schedule to review this paper, and we have carefully revised it in accordance with your suggestions. The responses to your comments and questions are shown in the attachment.
Best regards,
Xiangyu Chen and Yasuhiro Takahashi

Reviewer 3 Report
Overall review: The authors dealt with TIA with floating active inductor. However, written sections are not enough to stress out the advantage of the FAI for TIA compared to other TIAs except area reduction. In addition, analyses and comparisons have to be changed and further conducted since the authors have no measurement results. Unfortunately, that’s why this paper is not suitable to MDPI Electronics. Detailed comments are written below.
1.Lack of novelty
The authors just simply introduced the existing topology of FAI to RGC-based TIA. To support their idea and novelty, the authors should explain why Mahmoudi and Salama active inductor scheme has been chosen among other FAI schemes. Also the authors should explain why they chose RGC topology instead of inverter-based topology.
2.Lack of analyses
It would be better to explain about input sensitivity which is one of the most important parts for TIA. In addition, as authors mentioned about drawbacks for noise part due to FAI, they should analyze noise and other trade-offs in detail.
3.Lack of comparison and measurement results
The authors only compared their work with other TIAs fabricated in 0.18-um technology node. I think recent publications can be further considered by normalizing area. Also, they can simulate different TIA topologies with the same condition and then compare the performance. This might be important since the paper is focused on giving new idea with no measurement results. Topology comparison will make their argument stronger.
4.Minor comments
For page 5, line 109: must be decreased --> increased
For page 8, line 131: for short-haul links, higher input-referred noise currents can be tolerated unlike in long-haul links --> I think dispersion can be problem but not input-referred noise. Please give the supporting details
Author Response
Dear reviewer
We appreciate your comments on this article and have carefully revised it in accordance with your suggestions. The responses to your comments and questions are shown in the attachment.
Best regards,
Xiangyu Chen and Yasuhiro Takahashi

Round 2
Reviewer 1 Report
Your improved work is good.
Author Response
Dear reviewer,
We greatly appreciate all the insightful comments you have given us, which have greatly helped us to improve the manuscript. We believe that this work will be of sufficient value to the scientific community, especially those groups and
researchers engaged in optical communication systems.
Kind Regards,
Xiangyu Chen and Yasuhiro Takahashi
Reviewer 2 Report
The authors answer all my concerns.
Author Response

(The authors gave the same response as above.)

Reviewer 3 Report
Overall review: Thanks for the response. Still, there are some parts unclear which should be clarified and discussed. Detailed comments are written below.
i) The authors made a response that they realized inductor L2 with the FAI scheme. However, FAI requires two ports with transistors for the implementation. Since the TIA is implemented in single-ended input/output as shown in Figure 8, how did you make two ports for FAI?
ii) For 2nd comments, the authors mentioned that the reference [1] didn’t include bond wire inductance. For my opinion, I think that the reference [1] predicted 0.77nH (~1mm length of bond wire) as bond wire inductance. Therefore, I think just inductance of the bond wire is quite different with [1].
iii) From ii), that’s why I don’t get the point with the noise part. I think the structures from your work is quite similar with the reference [1] except replacing L2 with active inductor. And the authors mentioned that L2 doesn’t impact the noise performance that much. However, I think that’s when the when L2 is implemented with lumped spiral inductor.
Can you give more analysis when implemented with active inductor? I think active inductor will impact and boost the noise performance since they are composed of transistors. For example, you can compare input-referred noise or jitter of the eye diagram with the case of spiral inductor. Please justify that you can ignore the noise from the active inductor.
iv) For Figure 11, is it just transient simulation? The authors should include with noise factor to see the noise effects.
v) I understand the authors put TIAs as references with the similar bandwidth. However, for the Table I, the authors didn’t normalize area for FoM. How about scaling with process node when including chip area? For example, you can normalize area of [15] with 0.0002mm2 * (180/40)^2.
Author Response
Dear reviewer,
Thank you very much for your insightful comments, which have greatly helped us to improve the manuscript. We have carefully revised the manuscript in accordance with your comments and responded to you for your questions in detail.
Kind Regards,
Xiangyu Chen and Yasuhiro Takahashi
